# Prognostic value of procalcitonin in acute exacerbation of chronic obstructive pulmonary disease: A systematic review and meta-analysis

**Pang Qiyuan[1], Lin Changyang[2], Li Gaigai[2], Qiu Ju[2], Zhou Xun [2]***

**1** Department of Nursing, Guiyang Hospital of Stomatology, Guiyang, China, **2** Department of Pulmonary and Critical Care Medicine, The Second Affiliated Hospital of Guizhou University of Traditional Chinese Medicine, Guiyang, China

☯ These authors contributed equally to this work.
* 3453730998@qq.com

**Data Availability Statement:** All relevant data are within the manuscript.

**Funding:** Open access funding provided by The Project of National Famous Traditional Chinese

## Abstract

### Aims

To evaluate the prognostic role of procalcitonin(PCT) on all-cause mortality in acute exacerbation of chronic obstructive pulmonary disease (AECOPD).

### Methods

Database including PubMed, EMBASE, and the Cochrane Central Register of Controlled Trials were searched to find relevant trials. We included studies with patients hospitalized for AECOPD, which assessed procalcitonin levels and reported on the association between procalcitonin and mortality..

### Results

Fourteen trials involving 2983 patients were included. We found PCT levels in patients hospitalized for AECOPD are not associated with increased risk of mortality (RR 1.03, 95%CI 0.99–1.08). However, subgroup analysis showed PCT levels are indeed associated with an increased risk of mortality in mild to moderate AECOPD(RR 1.74, 95%CI 1.11–2.73). Deceased patients had significantly higher PCT levels, compared to survivors (MD 0.61, 95%CI 0.30–0.92). In PCT positive group, there was a significant increase in all-cause mortality(OR 3.21, 95%CI 1.84–5.61).

### Conclusions

Results from this meta-analysis suggest that procalcitonin levels at the time of hospital admission for mild to moderate AECOPD are positively correlated with mortality.

Medicine Practitioner Zhusheng Zhu Studio
(No.2022-75.

**Competing interests:** The authors have declared
that no competing interests exist.

## Introduction

Chronic obstructive pulmonary disease (COPD) is a common, preventable, and treatable disease that characterized by persistent respiratory symptoms and irreversible airflow limitation [1, 2]. Globally, COPD is the third most common cause of mortality, contributing to 6% of total deaths in 2019, with a majority occurring in low- and middle-income nations [3, 4]. Acute exacerbations of chronic obstructive pulmonary disease (AECOPD) play a singnificant role in the management of COPD, due to its negative impacts on health status and disease progression [5]. Hospitalization for COPD exacerbation is linked to an unfavorable prognosis and an elevated mortality risk [6].

Numerous investigations have been conducted to ascertain the prognostic indicators of mortality in patients with acute exacerbations of chronic obstructive pulmonary disease (AECOPD). It appears that that several clinical factors are associated with mortality from AECOPD, such as age, lower body mass index(BMI), forced expiratory volume in one second (FEV1), cardiac failure, diabetes mellitus, ischemic heart disease, malignancy, long-term oxygen requirement, and partial pressure of oxygen (PaO2) upon admission [7, 8]. However, there is little information about the relationship between biomarkers and death in COPD. Given above, effective biomarkers are sought for determining the acute attack frequency, length of hospitalization, severity of morbidity, and mortality.

Procalcitonin(PCT) is a small protein that is normally undetectable in plasma, which is significantly increased in the bacterial infections [9]. It has been shown that PCT-guided antibiotic therapy can safely reduce antibiotic overuse in patients with AECOPD [10]. PCT has been reported to be related to mortality in pneumonia and severe sepsis [11, 12]. Therefore, we conducted a comprehensive systematic review and meta-analysis to assess the value of procalcitonin in predict mortality among patients hospitalized for AECOPD.

## Methods

The present study was performed in accordance with the Preferred Reporting Items for Systematic Reviews and Meta-analyses(PRISMA) statement (S1 Checklist) and the Meta-analysis of Observational Studies in Epidemiology(MOOSE) checklist (S1 File) [13, 14]. The protocol for this study has been published on PROSPERO (registration number CRD42020158430).

### Search strategy and selection criteria

We conducted a systematic search of the PubMed, EMBASE and Cochrane Library databases for articles published up to October 2023 in any language. The search strategies were as follows: (COPD or "chronic obstructive pulmonary disease" or "chronic airflow obstruction" or "chronic obstructive airway disease" or "chronic obstructive lung disease") and (procalcitonin or PCT or pro-calcitonin or "calcitonin precursor polyprotein" or "calcitonin 1" or "calcitonin related polypeptide alpha") (S1 Table). Additional data sources were examined, including conference proceedings and the reference lists of the relevant studies.

Studies fulfilling the following selection criteria were included in this meta-analysis: (1) the type of study design was observational research, (2) the relationship between procalcitonin and mortality in patients with AECOPD, (3) diagnosis of COPD based on guidelines from the Global Initiative for Chronic Obstructive Lung Disease (GOLD). Reviews, case reports, conference abstracts, and animal experiments were excluded. If multiple studies used the same patient sample, the most recent or informative article was included. Two reviewers (Lin C and Li G) independently performed the search strategy and evaluated the studies. Any disagreement was resolved by a third reviewer (Pang Q).

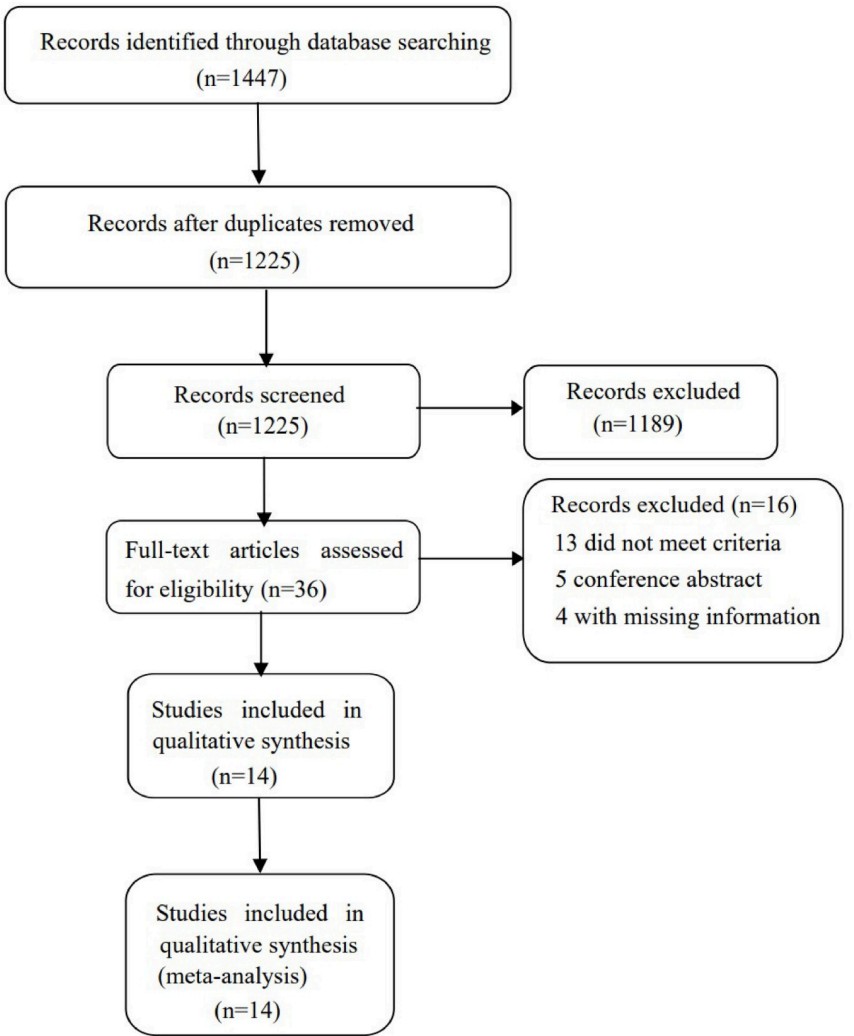

**Fig 1. Diagram illustrating the process for identifying relevant trials.**

## Study selection

The process of identifying relevant trials is shown in Fig 1. A search of databases revealed a total of 1447 studies, after removal of duplicates there were 1225 studies left for analysis. During the screening process, 1189 studies were eliminated for various reasons based on the title and abstract. Subsequently, fourteen studies were included in our analysis [15–26], and two [15, 23] of which used the same sample. All studies identified after excluding duplications and reasons for exclusion of each study is shown in S2 Table. Eight studies reported the ability of PCT predicts mortality in AECOPD. Five studies compared mortality according to PCT levels. And four studies compared PCT levels between survivors and deceased. Eleven studies were conducted in Europe, and three in Asia. The characteristics of the included trials are showed in Table 1. All data extracted in primary studies is shown in S3 Table.

## Date extraction

Two authors (Lin C and Li G) independently reviewed the full manuscripts of eligible trails, and the relevant data were extracted into predesigned data collection forms. We verified the

**Table 1. Characteristic of included trials associating with mortality.**

| Author | Year | Clinical Setting | Study design | Sample | Follow-up |
|---|---|---|---|---|---|
| **Stolz** | 2007 | ED | Prospective Study | 167 | Hospital/6 months |
| | 2008 | | | | 2 years |
| **Rammaert** | 2009 | ICU | Prospective Study | 116 | In-hospital |
| **Zuur-Telgen** | 2014 | Hospitalization | Prospective Study | 181 | 3 years |
| **Ceylan** | 2015 | Hospitalization | Not Mentioned | 58 | In-hospital |
| **Grolinund** | 2015 | Hospitalization | Prospective Study | 217 | 5–7 years |
| **Kutz** | 2015 | Hospitalization | Prospective Study | 584 | 30 days |
| **Ergan** | 2016 | ICU | Retrospective Study | 63 | In-hospital |
| **Flattet** | 2017 | Hospitalization | Retrospective Study | 359 | 5 years |
| **Gong** | 2020 | Hospitalization | Prospective Study | 110 | 6 months |
| **Yu** | 2020 | Hospitalization | Prospective Study | 695 | In-hospital |
| **Galani** | 2021 | ICU | Retrospective Study | 127 | 28 days |
| **Yao** | 2021 | Hospitalization | Retrospective Study | 146 | 28 days |
| **Koc** | 2022 | Hospitalization | Prospective Study | 160 | 6 months |

ED: emergency department

ICU: intensive care unit

accuracy of relevant data by comparing the collection forms. Any discrepancies between two reviewers were resolved through the evaluation of a third reviewer (Pang Q). The following data were collected from each study: author, year, study design, location, sample size, follow-up, all factors considered at multivariate analysis. Authors of included studies were contacted via email if further study details were needed.

## Qualitative assessment

To assess the quality of the eligible studies, two independent authors (Lin C and Li G) scored the studies according to the Newcastle-Ottawa Scale (NOS) for cohort studies. The 9-point NOS contains three items: selection (0–4), comparability (0–2), and exposure/outcome (0–3) (S4 Table). Studies that scored over 7 points on the NOS were deemed to be of high quality. Any disagreement was resolved via the evaluation of a third reviewer (Pang Q).

## Statistical analysis

For PCT predicts mortality, point estimates and standard errors were extracted from individual studies and were combined using the generic inverse variance method. The estimates of log ORs and standard errors have been obtained from the results of Cox proportional hazards regression models of the included studies. Subgroup analysis was performed based on the severity of disease. Severe cases are defined as requiring intubation or mechanical ventilation, and others as mild to moderate.

Continuous variables were pooled as mean difference (MD) and 95% confidence interval (CI) using inverse variance method. Mortality was analyzed using the Mantel-Haenszel (M-H) method to calculate odds ratio (OR) and 95% confidence interval (CI). The outcome measure was assessed in an intention-to-treat (ITT) manner. P-values<0.05 were considered statically significant. Considering the high likelihood of between-study variance, we used a random effect model. Heterogeneity among studies was assessed by Cochrane Q and $I^2$ statistics, $I^2$

values of more than 50% were considered to represent significant heterogeneity [27]. All data analyses were performed using the RevMan(v 5.3) software.

## Results

### PCT predict mortality

Eight clinical trials [15–22] with a total of 2085 patients reported that the level of PCT predicts mortality in patients with AECOPD. Five of these trials were prospective and three were retrospective. All trials showed that an elevated PCT level was associated with a higher risk of all-cause mortality, with hazard ratios (HR) ranging from 1.00 to 6.12. The pooled RR was 1.03 (95% CI, 0.99–1.08)(Fig 2). There was high heterogeneity across studies for this outcome, with $I^2$ = 79%(95% CI, 58%-89%).

Of these, five of the trials evaluated patients admitted to hospital or emergency department due to AECOPD, and the other three trials evaluated severe case of AECOPD requiring intubation and mechanical ventilation. We performed subgroup analysis based on the severity of AECOPD(Fig 3). The pooled RR for severe AECOPD was 1.01(95% CI, 0.97–1.05), with moderate heterogeneity($I^2$ 65%; 95% CI 0%-90%). The pooled RR for mild to moderate AECOPD was 1.74(95% CI, 1.11–2.73), with high heterogeneity($I^2$ 85%; 95% CI 67%-93%).

### Mortality in different levels of PCT

Five trials with 514 patients compared mortality between groups according to PCT levels. The PCT cutoff levels was 0.25ng/ml for four of the trials, and 0.10ng/ml for the remaining trial. There was a significant increase in all-cause mortality in PCT positive group (OR 3.21, 95% CI 1.84–5.61; $I^2$:0%) (Fig 4).

### Comparison of PCT levels between deceased and survivals

Four trials with 1566 patients compared PCT levels between survivors and deceased. The pooled data showed that deceased patients had significantly higher levels of PCT, compared to survivors (MD 0.61, 95% CI 0.30–0.92) (Fig 5). There was moderate heterogeneity across studies, with $I^2$ = 52%(95% CI, 0%-84%).

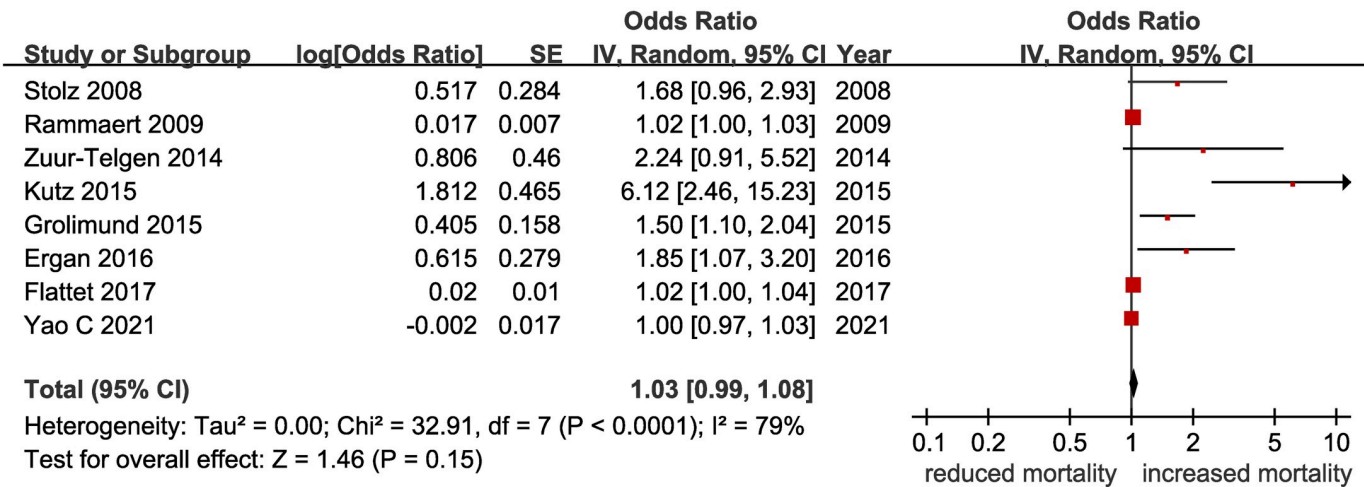

**Fig 2. The association between procalcitonin levels and mortality in patients with AECOPD.**

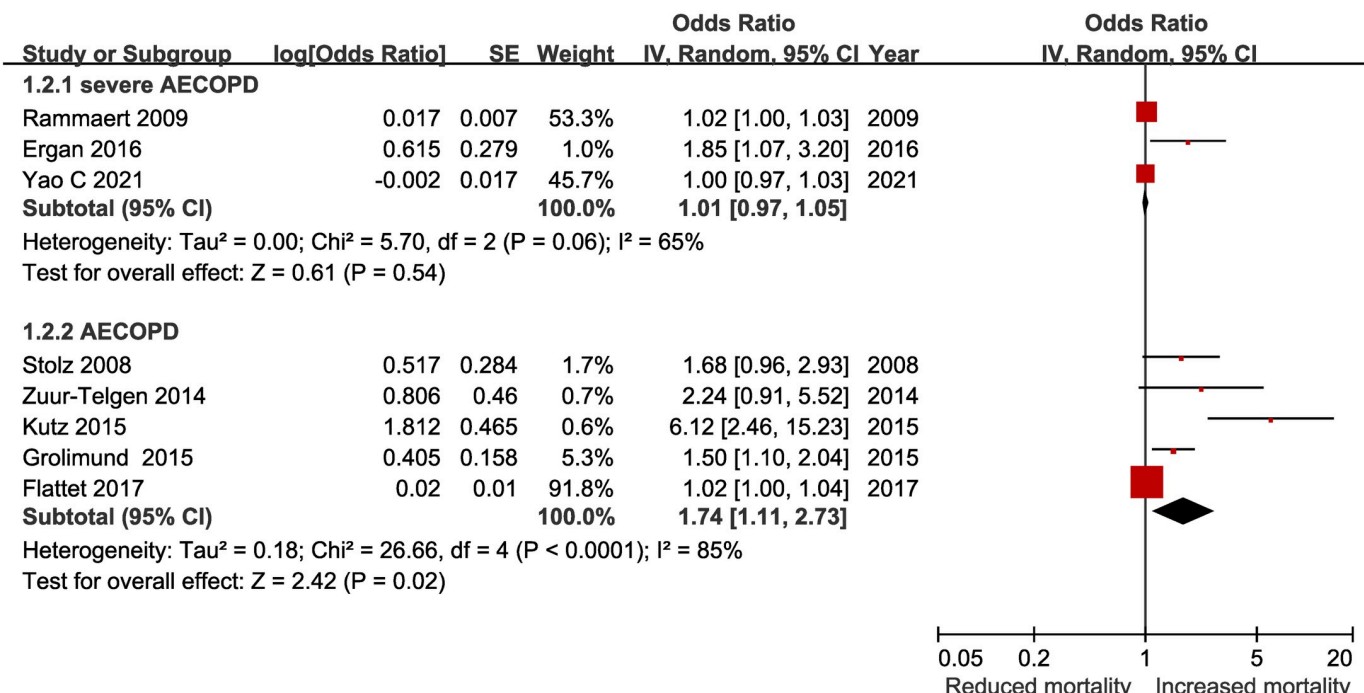

**Fig 3. The association between procalcitonin levels and mortality in patients with AECOPD stratified by the severity of disease.**

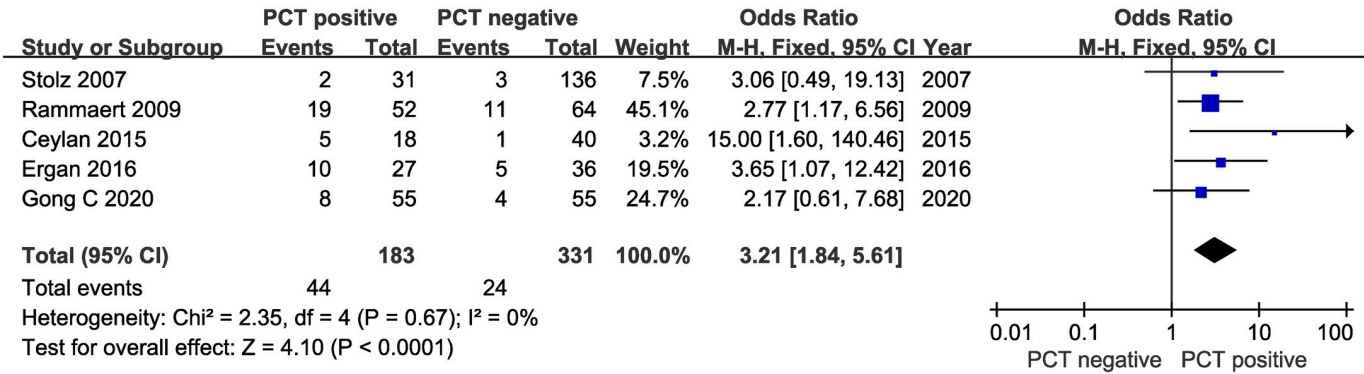

**Fig 4. Meta-analysis of studies compared mortality between groups according to PCT levels.**

## Discussion

In the current meta-analysis, the results determined that PCT levels in patients hospitalized for AECOPD are not associated with increased risk of mortality. However, we found that mortality is significantly higher in PCT positive group(P<0.05), and the PCT levels in deceased patients are significantly higher than those in survivors(P<0.05).

COPD is a prevalent chronic disease, with high mortality and morbidity. The frequency and severity of acute exacerbation are the most important factors determining overall prognosis in COPD, which both short- and long-term mortality rates are increased [28]. Therefore, early identification of high-risk patients for mortality is particularly important for improving

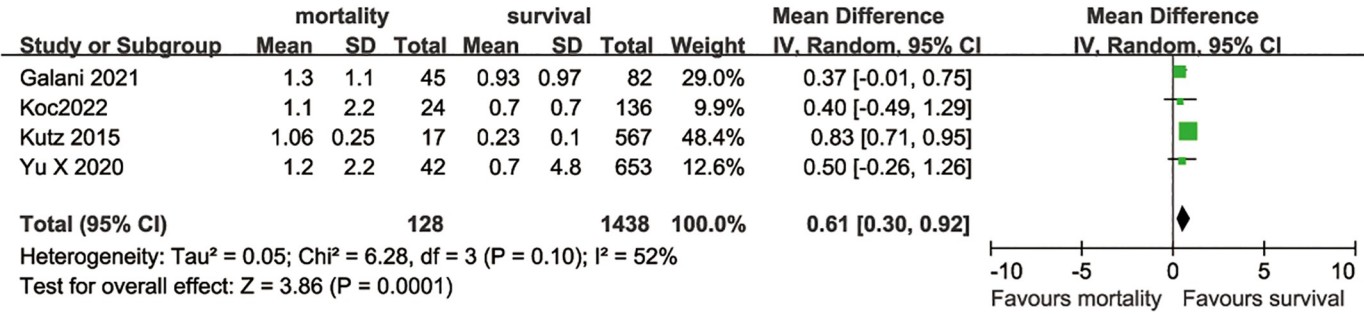

**Fig 5. Meta-analysis of studies compared PCT levels between survivors and deaths.**

prognosis. However, there is no single inflammatory marker used to predict the prognosis of AECOPD. Bacterial infection is a leading reason for AECOPD, and antibiotic treatment is necessary for the patients [29]. PCT is a specific marker of systemic bacterial infection and its levels correlate with the severity in critically ill patients [18, 30]. Several studies have suggested that the level of PCT in patients with AECOPD are higher than those in stable stage [31, 32]. Wang et al. [33] showed PCT levels are positively correlated with the severity of COPD. It is reasonable to assume that the level of PCT is associated with an increased risk of death. In our review, PCT levels are significantly higher in deaths of AECOPD compared to survivors. Correspondingly, the mortality rates were significantly increased in patients with high PCT level. However, according to the multivariate analysis of the included studies, PCT levels were not a useful predictive indicator of mortality in AECOPD. Subgroup analysis showed PCT levels are associated with an increased risk of mortality in patients with mild to moderate AECOPD.

There are several reasons that may explain these differences. First, It may be that because of PCT are predominantly increased in exacerbations caused by bacterial infections which is not the only factor leading to the exacerbation of COPD. In fact, only approximately 50% of exacerbations of COPD are associated with the isolation of bacteria from the lower respiratory tract [28, 34, 35]. Second, the analysis showed high statistical heterogeneity between studies ($I^2 = 79\%$), which may caused by differences in confounding factors, statistical analysis, determination of PCT and the definition of COPD. And the weights of included studies range from 1.0% to 53.3%. PCT also showed an association with the risk of death in severe AECOPD, when excluding study with the lowest weight. In addition, an individual patient data meta-analysis done by Kutz et al. showed that PCT level was a good indicator of mortality in patients with AECOPD, with an adjusted odds ratio of 6.12(95% CI: 2.46–15.18; p<0.001) [18]. In the future, the use of PCT should be further assessed in large, well-designed, prospective studies.

This study was subject to several limitations First, all included trials were conducted in Europe and Asia, thereby limiting the generalizability of the findings. Second, the multivariate analysis employed in various studies revealed substantial disparities in the range of adjusting factors incorporated, which may hinder the direct comparison of results. Finally, although our meta-analysis was solely reliant on published studies, it is acknowledged that publication bias remains a significant challenge, and potentially introduce a certain degree of bias in the overall analysis.

## Conclusions

Our meta-analysis shows that procalcitonin levels at the time of hospital admission for mild to moderate AECOPD are positively correlated with mortality.

## Supporting information

**S1 Checklist. PRISMA 2020 checklist.**
(DOCX)

**S1 File. MOOSE checklist.**
(PDF)

**S1 Table. Search strategy.**
(DOC)

**S2 Table. Studies identified after excluding duplications.**
(DOCX)

**S3 Table. Raw data used in current meta-analysis.**
(DOCX)

**S4 Table. Risk of bias for each study.**
(DOC)

## Author Contributions

**Conceptualization:** Pang Qiyuan, Zhou Xun.

**Data curation:** Lin Changyang, Li Gaigai, Qiu Ju.

**Formal analysis:** Pang Qiyuan, Lin Changyang.

**Methodology:** Lin Changyang, Li Gaigai, Qiu Ju.

**Software:** Qiu Ju.

**Writing – original draft:** Pang Qiyuan.

**Writing – review & editing:** Zhou Xun.

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
