## [Decision Letter · Decision Letter 0]

25 Jun 2024

PONE-D-24-13109Prognostic Value of Procalcitonin in Acute Exacerbation of Chronic Obstructive Pulmonary Disease: A Systematic Review and Meta-analysisPLOS ONE

Dear Dr. Xun,

Thank you for submitting your manuscript to PLOS ONE. After careful consideration, we feel that it has merit but does not fully meet PLOS ONE’s publication criteria as it currently stands. Therefore, we invite you to submit a revised version of the manuscript that addresses the points raised during the review process. The manuscript is within the scope of the journal. However, there are a few major concerns related to the manuscript that needs to be addressed before final decision.

Major comments:

Manuscript concludes that the deceased patients has significantly higher PCT, whereas SMD between two groups is not statistically different (SMD 0.94, 95%CI 0.74-1.14). This needs to be addressed. Also, the manuscript did not provide any information regarding publication bias (Funnel plot). Also, authors need to include the clinical implication of the findings of this metanalysis. There are a few other issues raised by reviewers.Manuscript also requires a through check for grammar and typo errors.

We look forward to receiving your revised manuscript.

Kind regards,

Vijay Hadda, MD

Academic Editor

PLOS ONE

Journal Requirements:

- https://doi.org/10.1371/journal.pone.0129450

- https://doi.org/10.1016/j.ijcard.2015.05.006

(among others)

In your revision ensure you cite all your sources (including your own works), and quote or rephrase any duplicated text outside the methods section. Further consideration is dependent on these concerns being addressed.

"Open access funding provided by The Project of National Famous Traditional Chinese Medicine Practitioner Zhusheng Zhu Studio (No.2022-75). "

“The authors received no specific funding for this work.”

5. We note that your Data Availability Statement is currently as follows: [All relevant data are within the manuscript and its Supporting Information files]

7. Please include your tables as part of your main manuscript and remove the individual files. Please note that supplementary tables (should remain/ be uploaded) as separate "supporting information" files

Reviewers' comments:

Reviewer's Responses to Questions

**Comments to the Author**

1. Is the manuscript technically sound, and do the data support the conclusions?

Reviewer #1: Partly

Reviewer #2: Partly

2. Has the statistical analysis been performed appropriately and rigorously? 

Reviewer #1: Yes

Reviewer #2: Yes

3. Have the authors made all data underlying the findings in their manuscript fully available?

Reviewer #1: Yes

Reviewer #2: Yes

4. Is the manuscript presented in an intelligible fashion and written in standard English?

Reviewer #1: Yes

Reviewer #2: Yes

5. Review Comments to the Author

Reviewer #1: In current metanalysis, the study question, study selection, data analysis and tests for heterogenity are performed as per standard protocol. However the studies included are quite heterogeneous and smaller studies or studies with less weight support the hypothesis better as compared to studies with higher weight. The funnel plot for heterogenity is not provided the authors. The actual cut off for PCT-Q for predicting the mortality is not mentioned. This metanaysis doesn't completely support the hypothesis.

Reviewer #2: The authors have highlighted the role of PCT at time of admission in predicting the mortality in patients with AECOPD. However, the following suggestions can be incorporated.

1. In the introduction (Line 51), grammatical errors are there. ‘Negatively impacts on health status’. It should be ‘negative impact on health’.

2. In the results, 8 clinical trials with 2085 patients are mentioned however, the authors have mentioned that fourteen clinical trials were taken into consideration for meta- analysis in the discussion. Pl clarify.

3. The discussion is not lucid as the authors have not clearly explained the results. The discussion meanders a bit and should be a bit more elaborate.

4. The authors have contradicted themselves. In the results, it is mentioned that no significant association between PCT levels and mortality was found. However, in the conclusion the authors have mentioned that PCT at the time of admission positively corelated with mortality. Please explain and clarify.

5. The Forest plots have not been labelled.

6. The authors have neither mentioned about the findings of sub- group analysis (as shown in the Forest plot) in the results nor have they discussed them in the discussion section.

6. PLOS authors have the option to publish the peer review history of their article (what does this mean?). If published, this will include your full peer review and any attached files.

Reviewer #1: **Yes: **Dr Neetu Jain

Reviewer #2: No

---

## [Author Response · Author response to Decision Letter 0]

2 Jul 2024

Dear Editors and Reviewers:

Thank you for your letter and for the reviewers’ comments concerning our manuscript entitled “Prognostic Value of Procalcitonin in Acute Exacerbation of Chronic Obstructive Pulmonary Disease: A Systematic Review and Meta-analysis” (ID: PONE-D-24-13109). Those comments are all valuable and very helpful for revising and improving our paper, as well as the important guiding significance to our researches. We have studied comments carefully and have made correction which we hope meet with approval. Revised portion are marked in red in the paper. The main corrections in the paper and the responds to the reviewer’s comments are as flowing:

Responds to the reviewer’s comments:

1. In the introduction (Line 51), grammatical errors are there. ‘Negatively impacts on health status’. It should be ‘negative impact on health’.

Response: We are very sorry for our negligence of grammar. And we have made modifications in the manuscript. To rectify the situation, we have carefully reviewed the entire document and implemented the necessary changes to correct the grammatical issues.

2. In the results, 8 clinical trials with 2085 patients are mentioned however, the authors have mentioned that fourteen clinical trials were taken into consideration for meta- analysis in the discussion. Pl clarify.

 Response: We deeply apologize for any misunderstandings caused by our expression. To comprehensively explore the correlation between PCT levels and mortality in AECOPD patients, we conducted this meta-analysis and included 14 high-quality studies. 

Of these, 8 studies explored the potential relationship between PCT levels and mortality risk in AECOPD patients by multivariate analysis, such as age, gender, underlying disease, etc. The results of these studies are usually presented in the form of risk ratio (HR) or odds ratio (OR). 

Five studies conducted comparative studies by PCT level grouping. In these studies, patients were divided into PCT positive and negative groups, and the mortality rates between the two groups were compared. This method can directly observe the impact of PCT levels on the survival status of patients.

In addition, four studies directly compared the PCT levels of deceased patients and survivors. These studies revealed the potential relationship between PCT levels and patient survival status by collecting and analyzing PCT data from two groups of patients. 

 In the results section of the meta-analysis, we analyzed different situations mentioned above. For the convenience of reading and understanding, we have made revisions in the results section. Special thanks to you for your good comments. 

3. The discussion is not lucid as the authors have not clearly explained the results. The discussion meanders a bit and should be a bit more elaborate.

Response: We have made correction according to the Reviewer’s comments.

4.The authors have contradicted themselves. In the results, it is mentioned that no significant association between PCT levels and mortality was found. However, in the conclusion the authors have mentioned that PCT at the time of admission positively corelated with mortality. Please explain and clarify.

Response: We have re-written the result and discussion sections according to the Reviewer’s suggestion. In our meta-analysis, the results showed that mortality is significantly higher in PCT positive group, and the PCT levels of deceased patients are significantly higher than those of survivors. 

However, the results determined that PCT levels in patients admitted to the hospital for AECOPD are not associated with increased risk of mortality(RR 1.03, 95%CI 0.99-1.08). Subgroup analysis showed PCT levels are associated with an increased risk of mortality in mild to moderate AECOPD(RR 1.74, 95%CI 1.11-2.73). 

Therefore, we have changed the conclusion section to”Results from this meta-analysis suggest that procalcitonin at the time of hospital admission for mild to moderate AECOPD is positively correlated with mortality”. 

5. The Forest plots have not been labelled.

Response: We have bolded the font in the citation section, and inserted figure captions in the text of the manuscript, immediately following the paragraph in which the figure is first cited.

6. The authors have neither mentioned about the findings of sub-group analysis (as shown in the Forest plot) in the results nor have they discussed them in the discussion section.

Response: It is really true as Reviewer suggested that the absence of subgroup analysis in our study is a notable limitation. Therefore, we have conducted subgroup analysis based on severity of the condition in our revised manuscript.

We tried our best to improve the manuscript and made some changes in the manuscript. These changes will not influence the content and framework of the paper. And here we did not list the changes but marked in red in revised paper.

We appreciate for Editors/Reviewers’ warm work earnestly, and hope that the correction will meet with approval.

Once again, thank you very much for your comments and suggestions.

---

## [Decision Letter · Decision Letter 1]

23 Aug 2024

PONE-D-24-13109R1Prognostic Value of Procalcitonin in Acute Exacerbation of Chronic Obstructive Pulmonary Disease: A Systematic Review and Meta-analysisPLOS ONE

Dear Dr. Xun,

Thank you for submitting your manuscript to PLOS ONE. After careful consideration, we feel that it has merit but does not fully meet PLOS ONE’s publication criteria as it currently stands. Therefore, we invite you to submit a revised version of the manuscript that addresses the points raised during the review process.  Academic Editor's comments: 1. Line 81-82, mention “ the type of 82 study design was observational research”, table mention study design retrospective, prospective cohort or cohort study. Line 131 – 132 mention “Eight clinical trials [14-21] with a total of 2085 patients reported the level of PCT predicting mortality for patients with AECOPD”

It is confusing; authors must use uniform and standard terminology. RCT and cohort study are two different study designs.

2. Please clarify and modify in table whether clinical setting “Hospital” represent out patient department or hospitalized or both? Mild exacerbations usually donot require hospitalization. Hence, it will be good to include more clear information where the study was conducted – out patient department or hospitalized or both. This is essential, as authors concluded that PCT was correlated with increased mortality in mild to moderate.

3. Authors need to review for typo-errors

We look forward to receiving your revised manuscript.

Kind regards,

Vijay Hadda, MD

Academic Editor

PLOS ONE

Journal Requirements:

Reviewers' comments:

Reviewer's Responses to Questions

**Comments to the Author**

1. If the authors have adequately addressed your comments raised in a previous round of review and you feel that this manuscript is now acceptable for publication, you may indicate that here to bypass the “Comments to the Author” section, enter your conflict of interest statement in the “Confidential to Editor” section, and submit your "Accept" recommendation.

Reviewer #1: (No Response)

Reviewer #2: All comments have been addressed

2. Is the manuscript technically sound, and do the data support the conclusions?

Reviewer #1: Partly

Reviewer #2: Yes

3. Has the statistical analysis been performed appropriately and rigorously? 

Reviewer #1: Yes

Reviewer #2: Yes

4. Have the authors made all data underlying the findings in their manuscript fully available?

Reviewer #1: Yes

Reviewer #2: Yes

5. Is the manuscript presented in an intelligible fashion and written in standard English?

Reviewer #1: Yes

Reviewer #2: Yes

6. Review Comments to the Author

Reviewer #1: 1. It is still not clear how 14 studies were selected and results are reported only for 8 studies

2. Conclusions should not be based on only on subgroup analysis but also included the whole analysis and language need revision and should mention if it statistically significant relationship.

3. The results stating that PCT levels are associated with high mortality in mild to moderate COPD are contradictory as mild to moderate COPD by definition doesn't even require hospitalization.

4. Discussion needs revision and should explain the results better.

5. There are many grammatical errors like Line 131-132 "Eight clinical trials [14-21] with a total of 2085 patients reported the level of PCT predicting mortality for patients with AECOPD " should be "Eight clinical trials [14-21] with a total of 2085 patients reported that the level of PCT predicts mortality in patients with AECOPD"

Reviewer #2: The authors have made significant changes in the MS and it has shaped up well.

The authors have mentioned in discussion section that COPD is a 'prevalent diagnosed chronic disease'. Please clarify what this implies?

7. PLOS authors have the option to publish the peer review history of their article (what does this mean?). If published, this will include your full peer review and any attached files.

Reviewer #1: **Yes: **Dr Neetu Jain

Reviewer #2: No

---

## [Author Response · Author response to Decision Letter 1]

29 Aug 2024

Dear Editors and Reviewers:

We quite appreciate your favorite consideration and the reviewer’s insightful comments concerning our manuscript entitled “Prognostic Value of Procalcitonin in Acute Exacerbation of Chronic Obstructive Pulmonary Disease: A Systematic Review and Meta-analysis” (ID: PONE-D-24-13109R1). Now We have revised the manuscript exactly according to the reviewer’s comments, and found these comments are very helpful. We hope this revision can make my paper more acceptable. The revisions were addressed point by point below.

Responds to the reviewer’s comments:

1.Line 81-82, mention “ the type of 82 study design was observational research”, table mention study design retrospective, prospective cohort or cohort study. Line 131 – 132 mention “Eight clinical trials [14-21] with a total of 2085 patients reported the level of PCT predicting mortality for patients with AECOPD”

Response: We are very sorry for our lack of rigor. And we have made modifications in the manuscript. 

2. Please clarify and modify in table whether clinical setting “Hospital” represent out patient department or hospitalized or both? Mild exacerbations usually donot require hospitalization. Hence, it will be good to include more clear information where the study was conducted – out patient department or hospitalized or both. This is essential, as authors concluded that PCT was correlated with increased mortality in mild to moderate.

 Response: We deeply apologize for such a misunderstanding. All articles in our current meta-analysis included patients hospitalized for AECOPD. For the convenience of reading and understanding, we have made revisions in the table. 

3.Authors need to review for typo-errors

Response: We have made revisions to the entire text according to the Reviewer’s comments.

In all, we found the reviewer’s comments are quite helpful, and revised my paper point-by-point. Thank you and the review again for your help!

---

## [Decision Letter · Decision Letter 2]

24 Sep 2024

PONE-D-24-13109R2Prognostic Value of Procalcitonin in Acute Exacerbation of Chronic Obstructive Pulmonary Disease: A Systematic Review and Meta-analysisPLOS ONE

Dear Dr. Xun,

Thank you for submitting your manuscript to PLOS ONE. After careful consideration, we feel that it has merit but does not fully meet PLOS ONE’s publication criteria as it currently stands. Therefore, we invite you to submit a revised version of the manuscript that addresses the points raised during the review process.

Please include the MOOSE checklist, mention how the publication bias was assessed/addressed, and the method section section needs more information (please check the comments of reviewer 3).

We look forward to receiving your revised manuscript.

Kind regards,

Vijay Hadda, MD

Academic Editor

PLOS ONE

Additional Editor Comments:

There are certain important issues related the statistical part of the manuscript. Please refer to the comments from the reviewer provided below.

Reviewers' comments:

Reviewer's Responses to Questions

**Comments to the Author**

1. If the authors have adequately addressed your comments raised in a previous round of review and you feel that this manuscript is now acceptable for publication, you may indicate that here to bypass the “Comments to the Author” section, enter your conflict of interest statement in the “Confidential to Editor” section, and submit your "Accept" recommendation.

Reviewer #1: All comments have been addressed

Reviewer #3: All comments have been addressed

2. Is the manuscript technically sound, and do the data support the conclusions?

Reviewer #1: Yes

Reviewer #3: Yes

3. Has the statistical analysis been performed appropriately and rigorously? 

Reviewer #1: Yes

Reviewer #3: Yes

4. Have the authors made all data underlying the findings in their manuscript fully available?

Reviewer #1: Yes

Reviewer #3: Yes

5. Is the manuscript presented in an intelligible fashion and written in standard English?

Reviewer #1: (No Response)

Reviewer #3: Yes

6. Review Comments to the Author

Reviewer #1: line 33 (abstract) And deceased patients had 1-" remove And before the sentence

" line 50 pulmonary disease (AECOPD) play a singnificant role in the management of COPD, due to their " replace "their" with "its"

" line 167 (dicussion) please mention p-value when word "significant" is used

Reviewer #3: As a statistical reviewer I will focus on methods and reporting.

Major

1) MOOSE is a checklist for reporting meta analyses. The authors need to assess the risk of bias in the studies using an appropriate tool (see Cochrane ROBINS for example). they also say "The reviewers assessed the quality of the included studies according to Modifying the MOOSE item list." this needs to be corrected for grammar and also they need to explain why they had to use a modified version and who modified it.

2) In the statitical analyses section, list all the outcomes, primary and secondary, for clarity. Also are all outcomes reported in exactly the same scale, so there is no need for meta analysing as SMDs? please state that clearly as well. UPDATE: I see in the results SMDs are reported, that needs to be explained in the methods section.

3) Publication bias is one of the biggest risks in conducting meta analyses and it seems to be ignored. Publication bias tests and plots only relevant if you have >10 studies otherwise underpowered to detect much and tend to lead to conclusions that are not justified http://www.ncbi.nlm.nih.gov/pubmed/11106885. If you don’t have enough studies to assess you should discuss this as a major limitation. Even with 10 or 20 studies it is very difficult to visually assess. If you have 20 or more studies it is a considerable strength.

4) Avoid fixed effect models since they under-perform in the presence of ANY heterogeneity. Random-effects (RE) models are more conservative and provide better estimates with wider confidence intervals: http://www.ncbi.nlm.nih.gov/pubmed/11252006 and http://www.ncbi.nlm.nih.gov/pubmed/21148194 . What does the arbitrary 50% cut-off point add? Why is 49% fine and 51% an issue? My point is that a RE model will work better, in the presence of 5% heterogeneity, compared to a FE model!

5) Report the confidence intervals for I^2, since it is an estimate. A simple formula exists in the seminal 2002 Higgins paper that proposed I^2.

Minor

1) "Subgroup analysis was performed based on the severity of disease" please expand to clarify, what were the grouping for example.

2) Year may be worth considering in bias assessment, especially if you don't have enough studies for a formal test: http://www.ncbi.nlm.nih.gov/pubmed/25988604. With newer studies we would be more confident.

3) How was the random-effect model implemented, i.e. how was heterogeneity estimated? There are numerous ways to do so. Did they use the standard DerSimonian-Laird method? If so, please state so. Also there are better performing methods, for example please see https://www.ncbi.nlm.nih.gov/pubmed/28815652 (or http://www.ncbi.nlm.nih.gov/pubmed/23922860) and the metaan command in Stata where these are implemented (https://www.stata-journal.com/article.html?article=st0201).

7. PLOS authors have the option to publish the peer review history of their article (what does this mean?). If published, this will include your full peer review and any attached files.

Reviewer #1: **Yes: **Dr Neetu Jain

Reviewer #3: No

---

## [Author Response · Author response to Decision Letter 2]

26 Sep 2024

Dear Editors and Reviewers:

We quite appreciate your favorite consideration and the reviewer’s insightful comments concerning our manuscript entitled “Prognostic Value of Procalcitonin in Acute Exacerbation of Chronic Obstructive Pulmonary Disease: A Systematic Review and Meta-analysis” (ID: PONE-D-24-13109R1). Now We have revised the manuscript exactly according to the reviewer’s comments, and found these comments are very helpful. We hope this revision can make my paper more acceptable. The revisions were addressed point by point below.

Responds to the reviewer’s comments:

1.line 33 (abstract) And deceased patients had 1-" remove And before the sentence,"line 50 pulmonary disease (AECOPD) play a singnificant role in the management of COPD, due to their " replace "their" with "its","line 167 (dicussion) please mention p-value when word "significant" is used

Response: We are very sorry for our lack of rigor. And we have made modifications in the manuscript. 

2. MOOSE is a checklist for reporting meta analyses. The authors need to assess the risk of bias in the studies using an appropriate tool (see Cochrane ROBINS for example). they also say "The reviewers assessed the quality of the included studies according to Modifying the MOOSE item list." this needs to be corrected for grammar and also they need to explain why they had to use a modified version and who modified it.

 Response: Our meta-analysis was performed in accordance with the Preferred Reporting Items for Systematic Reviews and Meta-analyses(PRISMA) statement. According to your comments, we have added the MOOSE checklist. 

3.In the statitical analyses section, list all the outcomes, primary and secondary, for clarity. Also are all outcomes reported in exactly the same scale, so there is no need for meta analysing as SMDs? please state that clearly as well. UPDATE: I see in the results SMDs are reported, that needs to be explained in the methods section.

Response: We have made correction according to the Reviewer’s comments, including the “Study selection” and “Statistical analysis” sections.

4.Publication bias is one of the biggest risks in conducting meta analyses and it seems to be ignored. Publication bias tests and plots only relevant if you have >10 studies otherwise underpowered to detect much and tend to lead to conclusions that are not justified http://www.ncbi.nlm.nih.gov/pubmed/11106885. If you don’t have enough studies to assess you should discuss this as a major limitation. Even with 10 or 20 studies it is very difficult to visually assess. If you have 20 or more studies it is a considerable strength.

Response: Thank you for your precious opinions, we have removed the funnel plot and discussed the publication bias. 

5.Avoid fixed effect models since they under-perform in the presence of ANY heterogeneity. Random-effects (RE) models are more conservative and provide better estimates with wider confidence intervals. What does the arbitrary 50% cut-off point add? Why is 49% fine and 51% an issue? My point is that a RE model will work better, in the presence of 5% heterogeneity, compared to a FE model!

Response: We have made correction according to the Reviewer’s comments. However, we still choose the fixed effects model when there is no heterogeneity present(I2=0).(Figure 4)

6.Report the confidence intervals for I2, since it is an estimate. A simple formula exists in the seminal 2002 Higgins paper that proposed I2.

Response: We have reported the confidence intervals for I2 according to the Reviewer’s comments.

7. "Subgroup analysis was performed based on the severity of disease" please expand to clarify, what were the grouping for example.

 Response: We have modified and clarified in “Statistical analysis” section. “Subgroup analysis was performed based on the severity of disease. Severe cases are defined as requiring intubation or mechanical ventilation, and others as mild to moderate.”

8.Year may be worth considering in bias assessment, especially if you don't have enough studies for a formal test. With newer studies we would be more confident.

Response: Thanks for your comments, we also agree that year is worth considering in bias assessment. However, there has been no significant change in the diagnostic criteria for COPD, and PCT and death are objective indicators. We believe that it is not a determining factor in bias assessment.

9.How was the random-effect model implemented, i.e. how was heterogeneity estimated? There are numerous ways to do so. Did they use the standard DerSimonian-Laird method? If so, please state so.

Response: We conducted this meta-analysis using Review Manager Software 5.3. RevMan implements a version of random-effects meta-analysis that is described by DerSimonian and Laird method.

We appreciate for Editors/Reviewers’ warm work earnestly, and hope that the correction will meet with approval.

Once again, thank you very much for your comments and suggestions.

---

## [Decision Letter · Decision Letter 3]

2 Oct 2024

Prognostic Value of Procalcitonin in Acute Exacerbation of Chronic Obstructive Pulmonary Disease: A Systematic Review and Meta-analysis

PONE-D-24-13109R3

Dear Dr. Xun,

We’re pleased to inform you that your manuscript has been judged scientifically suitable for publication and will be formally accepted for publication once it meets all outstanding technical requirements.

Kind regards,

Vijay Hadda, MD

Academic Editor

PLOS ONE

Additional Editor Comments (optional):

Dear authors,

Thank you for addressing all reviewers comments satisfactorily. The manuscript has improved significantly.

Reviewers' comments:

Reviewer's Responses to Questions

**Comments to the Author**

1. If the authors have adequately addressed your comments raised in a previous round of review and you feel that this manuscript is now acceptable for publication, you may indicate that here to bypass the “Comments to the Author” section, enter your conflict of interest statement in the “Confidential to Editor” section, and submit your "Accept" recommendation.

Reviewer #3: All comments have been addressed

2. Is the manuscript technically sound, and do the data support the conclusions?

Reviewer #3: Yes

3. Has the statistical analysis been performed appropriately and rigorously? 

Reviewer #3: Yes

4. Have the authors made all data underlying the findings in their manuscript fully available?

Reviewer #3: Yes

5. Is the manuscript presented in an intelligible fashion and written in standard English?

Reviewer #3: Yes

6. Review Comments to the Author

Reviewer #3: I am satisfied with the authors' responses and the resulting changes to the paper - some of the responses were difficult to follow (e.g. about MOOSE and the related action) but the paper has been amended satisfactorily.

7. PLOS authors have the option to publish the peer review history of their article (what does this mean?). If published, this will include your full peer review and any attached files.

Reviewer #3: No

---

## [Editor Report · Acceptance letter]

25 Oct 2024

PONE-D-24-13109R3 

PLOS ONE

Dear Dr. Xun, 

I'm pleased to inform you that your manuscript has been deemed suitable for publication in PLOS ONE. Congratulations! Your manuscript is now being handed over to our production team.

Kind regards, 

on behalf of

Dr. Vijay Hadda 

Academic Editor

PLOS ONE